# The Role of Tocotrienol in Protecting Against Metabolic Diseases

**DOI:** 10.3390/molecules24050923

**Published:** 2019-03-06

**Authors:** Kok-Lun Pang, Kok-Yong Chin

**Affiliations:** 1School of Pharmacy, University of Reading Malaysia, Iskandar Puteri Johor 79200, Malaysia; k.l.pang@reading.edu.my; 2Department of Pharmacology, Faculty of Medicine, Universiti Kebangsaan Malaysia, Kuala Lumpur 56000, Malaysia

**Keywords:** adipose, diabetes, insulin resistance, metabolic syndrome, obesity, overweight, vitamin E

## Abstract

Obesity is a major risk factor for diabetes, and these two metabolic conditions cause significant healthcare burden worldwide. Chronic inflammation and increased oxidative stress due to exposure of cells to excess nutrients in obesity may trigger insulin resistance and pancreatic β-cell dysfunction. Tocotrienol, as a functional food component with anti-inflammatory, antioxidant, and cell signaling-mediating effects, may be a potential agent to complement the current management of obesity and diabetes. The review aimed to summarize the current evidence on the anti-obesity and antidiabetic effects of tocotrienol. Previous studies showed that tocotrienol could suppress adipogenesis and, subsequently, reduce body weight and fat mass in animals. This was achieved by regulating pathways of lipid metabolism and fatty acid biosynthesis. It could also reduce the expression of transcription factors regulating adipogenesis and increase apoptosis of adipocytes. In diabetic models, tocotrienol was shown to improve glucose homeostasis. Activation of peroxisome proliferator-activated receptors was suggested to be responsible for these effects. Tocotrienol also prevented multiple systemic complications due to obesity and diabetes in animal models through suppression of inflammation and oxidative stress. Several clinical trials have been conducted to validate the antidiabetic of tocotrienol, but the results were heterogeneous. There is no evidence showing the anti-obesity effects of tocotrienol in humans. Considering the limitations of the current studies, tocotrienol has the potential to be a functional food component to aid in the management of patients with obesity and diabetes.

## 1. Introduction

Obesity is a multifactorial chronic medical condition affecting both children and adults worldwide. Clinically, the World Health Organization defines overweight as a body mass index (BMI) ≥ 25 kg/m^2^, and obesity as a BMI ≥ 30 kg/m^2^ in adults [1]. An estimate performed in 2015 found that 107.7 (98.7–118.7) million children and 603.7 (588.2–619.8) million adults suffered from obesity [2]. The abnormal or excessive fat accumulation in obese individuals poses adverse metabolic effects on their blood pressure, cholesterol, triglycerides levels, and glucose homeostasis [1,3]. Approximately 7.2% (4.9–9.4%) of all deaths and 4.9% (3.5–6.4%) of disability-adjusted-life-years (DALYs) worldwide in 2015 were attributable to excess body weight [2]. Of all deaths related to cardiovascular diseases, 41% was contributed by high BMI [2].

Obesity is a major risk factor of type 2 diabetes mellitus (T2DM). T2DM affects 463 million (424–509 million) individuals worldwide and causes 37.6 million (255–520 million) years-lived-with-disability in 2017 [4]. Uncontrolled T2DM can lead to complications in other organ systems, including neuropathy, nephropathy, oculopathy, and cardiovascular diseases [5]. The exact mechanism that links obesity to T2DM is not clearly understood at this moment, but the exposure of cells towards concentration of nutrients exceeding energy requirement is suggested to play a role. This condition initiates the recruitment and activation of macrophages and other immune cells and invokes tissue inflammation. Concurrently, a high concentration of nutrients also causes mitochondrial dysfunction characterized by reduced mitochondrial mass and function. Both processes subsequently lead to insulin resistance in liver, skeletal muscle, and adipose tissue, as well as pancreatic β-cell dysfunction. The latter causes a drop in insulin production, which elevates circulatory glucose concentration, further worsening the problem of excessive nutrients [6]. The chronic inflammatory state and mitochondrial dysfunction also elevated oxidative stress in obese individuals, leading to impaired β-cell regeneration and function [7,8].

The management of obesity involves behaviour and lifestyle modifications, including reducing calorie intake and increasing physical activity, to correct the calorie imbalance. Some medications, such as phentermine, lorcaserin, and orlistat, have been used to aid weight loss but they come with side effects. The effectiveness of bariatric surgery in weight management is gaining recognition, and it is the treatment of choice when the former methods fail [9]. The management of T2DM mainly involves the use of metformin and other hypoglycemic agents (sulfonylurea, pioglitazone, and dipeptidyl peptidase 4 inhibitor) to improve glycemic control [10]. For patients with obesity and T2DM, certain hypoglycemic agents may offset their efforts to reduce weight [11]. An agent that can help to manage obesity and T2DM concurrently would be very useful to patients suffering from both conditions.

Tocotrienols (T3s) are among novel compounds studied extensively for their metabolic effects. Being in the vitamin E (tocochromanols) family, they share a similar structure with tocopherols, consisting of a 6-chromanol ring and a hydrophobic carbon side chain. T3s differ from tocopherols by the presence of three unsaturated bonds at the positions of 3, 7, and 11 of the side chains. This explains the better incorporation of T3s in the lipid membrane and intermembrane transfer. T3s can be further divided into α-, β-, γ-, and δ-isoforms based on the position of number and position of the methyl group at the chromanol ring [12,13]. The chemical structure of all tocopherol and T3 isoforms is shown in Figure 1. Both T3s and tocopherols are found in natural botanical sources in a mixture of varying composition as a natural defence mechanism against lipid peroxidation [12]. Some of the natural sources of T3 include annatto oil (34.27–36.25 mg/mL of the lipid extract from the seeds), palm oil (738 mg/mL), rice bran oil, RBO (586 mg/mL), and wheat germ (26 mg/mL) [14,15].

Many studies have established the role of T3 in protecting against metabolic, diabetic and cardiovascular pathologies [16,17]. Both animal and human studies report that T3 supplementation improves glucose homeostasis [18]. A limited number of animal studies suggested that T3 reduces body weight or body fat [19,20]. As a free radical scavenger, T3 can ameliorate oxidative stress in metabolic disorders and protect cellular functions [21]. T3 can suppress cholesterol synthesis by inhibiting 3-hydroxy-3-methyl-glutaryl-coenzyme A reductase (HMGCR) post-transcriptionally, the rate-determining enzyme in the mevalonate pathway [22]. It has been shown to downregulate peroxisome proliferator-activated receptor γ (PPARγ)—the transcription factor critical in adipocytes differentiation [23]. It can prevent the activation of nuclear factor-κB (NF-κB), thereby halting tissue inflammation [24]. These biological activities suggest that T3 can be utilized in the management of metabolic conditions like obesity and T2DM. The objective of this review is to summarize current evidence on anti-obesity and antidiabetic effects of T3 derived from cellular, animal, and human studies.

## 2. Literature Search

The literature search was performed between the 1st and 25th May 2018 on PubMed and Scopus using the keywords “Obesity” or “Diabetes” and “Tocotrienol”. We also examined the reference lists of the retrieved articles. Original research articles regarding the beneficial effects of T3 and published in English were included. Studies of chemically modified T3 were excluded from this review. A total of 52 studies were included in the review. 

## 3. Tocotrienol and Obesity

### 3.1. Anti-Obesity Properties of T3

Adipogenesis refers to the differentiation of preadipocytes into mature adipocytes with increased synthesis and accumulation of intracellular triglyceride (TG) and lipid droplets [25,26]. T3 was reported to exert anti-obesity properties on several cell lines [27,28,29]. It was demonstrated to suppress adipogenesis by reducing intracellular TG and lipid droplet accumulation in mouse hepatoma Hepa 1-6 cells [30], human hepatoma HepG2 cells [20,31], 3T3-L1 preadipocytes [27,32], 3T3-F442A murine preadipocytes [23], and primary human adipose-derived stem cells (hASCs) [28,33] via oil red O staining assay. The potency of anti-adipogenic activities of T3 follows the order of γ-T3 > δ-T3 > β-T3 > α-T3 [27,28].

T3 was reported to reduce body weight, especially fat mass in several obese animal models [19,20,34,35,36]. A 20-week supplementation of rice bran enzymatic extract (RBEE) (5%) containing 174 mg/kg T3 reduced body weight gain, visceral abdominal adipose tissue weight, and adipocyte size distribution in obese Zucker rats [34]. Similarly, oral administration of tocotrienol-rich fraction (TRF), with mixed T3 and tocopherol (120 mg/kg body weight, bw/day), also significantly reduced total abdominal omental fat pads in high fat diet-fed rats [36]. Oral γ-T3 supplementation (60 mg/kg) also significantly suppressed dexamethasone-induced obesity with lower body fat mass on adrenalectomized rats [37]. Besides, γ-T3 (0.05% in diet) also normalized high fat diet-induced weight gain by reducing adipose tissues gain in the epididymal fat pad, mesenteric fat pad, and liver of C57BL/6J mice [35]. In contrast, Wong et al. reported that δ-T3 (85 mg/kg bw/day), but not α-T3 and γ-T3, significantly reduced total fat mass, abdominal circumference, adiposity index, and retroperitoneal and epididymal fat pads mass in high carbohydrate/fat diet-fed rats [19]. On the other hand, T3 treatment (60 mg/kg/day) by gastric gavage marginally increased the mesenteric adipose tissues deposition in dexamethasone-treated adrenalectomized rats [38]. This discrepancy may be partly due to the composition or purity of T3 used. In addition, Tocomin^®^ (palm tocotrienol mixtures, predominantly γ-T3) [21,39] and annatto oil T3 mixture (90% δ-T3 and 10% γ-T3) [40] were reported to have marginal effects on body weight and fat pad mass gain in high fat diet-fed rodents. To our best knowledge, there is no human study reported on the effect of T3 in preventing or reducing obesity.

### 3.2. Molecular Mechanism of Anti-Adipogenic Properties of T3

#### 3.2.1. Suppression of Adipogenesis by T3

Mechanistically, T3 (TRF, δ-T3, γ-T3, or β-T3) was demonstrated to inhibit adipogenesis by upregulating the lipid metabolism proteins/enzymes including carnitine palmitoyltransferase 1A/2 (CPT1A/2), Forkhead box A2 (FOXA2), cytochrome P450 3A4, adiponectin receptor protein 2 (ADIPOR2), uncoupling protein 2 (UCP2), and peroxisome proliferator-activated receptor α/δ (PPARα/δ) [20,27,30,40]. At the same time, T3 was also shown to downregulate fatty acid biosynthesis proteins/enzymes, including fatty acid synthase (FAS), sterol regulatory element-binding protein-1/2 (Srebf1/2), stearoyl-CoA desaturase-1 (Scd1), acetyl-CoA carboxylase-1 (ACC1), HMGCR, apolipoprotein, low-density lipoprotein receptor (LDLR), diglyceride acyltransferase (Dgat2), pyruvate kinase, and carbohydrate-responsive element-binding protein (ChREBP) [20,27,30,31,33,40]. 

Studies also showed that T3 inhibited adipogenesis via downregulation of adipogenic transcription factors expression and/or signaling, including Akt [23,32], CCAAT-enhancer-binding protein α (C/EBPα) [28,32,33] and PPARγ signaling [27,28,29,32,33]. Besides, T3 was also demonstrated to downregulate PPARγ downstream genes expression including fatty acid binding protein (FABP), adiponectin, glucose transporter type 4 (Glut4), perilipin, and hormone-sensitive lipase (HSL) [23,28,32,33]. In addition, TRF and γ-T3 (but not α-T3) significantly suppressed FABP and C/EBPα mRNA expression upon insulin stimulation in 3T3-L1 preadipocytes [32]. TRF (1 µM), α-T3 (0.24 µM), and γ-T3 (0.024 µM) were also reported to reduce insulin-mediated upregulation of PPARγ mRNA level during the differentiation of 3T3-L1 preadipocytes [32]. γ-T3 but not α-T3 significantly suppressed insulin-mediated PPARγ protein expression, Akt activation, and adipogenesis in 3T3-L1 preadipocytes [32]. Another similar study also revealed that δ-T3 suppressed adipogenesis of 3T3-F442A preadipocytes with the reduction of HMGCR expression and suppression of Akt and PPARγ signaling [23]. Molecular experiment further revealed that the antiadipogenic effect of δ-T3 was independent of HMGCR inhibition [23]. Rosiglitazone (PPARγ agonist), but not mevalonolactone (the product of HMGCR signaling), managed to revert δ-T3-suppressed preadipocyte differentiation [23]. Interestingly, Zhao et al. reported that γ-T3 and δ-T3 did not exert significant anti-adipogenic activities on the terminal and fully differentiated adipocytes [28]. Similarly, Matsunaga et al. also reported that γ-T3 restored tumor necrosis factor-α (TNF-α)-downregulated adiponectin and PPARγ mRNA levels in differentiated 3T3-L1 cells [41]. This may explain the inconsistent findings of T3-suppressed fat mass gain in different cells or obesity model.

#### 3.2.2. Pro-Apoptotic Effects of T3 on Adipocytes

On the other hand, T3 also exhibits anti-adipogenic activities by causing adipocyte cell death [27,28,29]. α-T3, γ-T3, and δ-T3 (10 and 20 µM) were reported to induce anti-adipogenesis via antiproliferation and apoptosis induction on 3T3-L1 preadipocytes [29]. Besides, all T3 isoforms were cytotoxic to 3T3-L1 preadipocytes at a concentration of 30–50 µM [27]. Furthermore, Zhao et al. reported that γ-T3 and δ-T3 but not α-T3 were cytotoxic to undifferentiated (5 µM) and differentiated (10 µM) primary hASCs [28]. The negative finding of α-T3 might be due to the much lower concentration used in this study. In addition, γ-T3 also induced autophagy in primary hASCs [28], however, it might not directly relate to the anti-adipogenic activity [28]. In apoptosis induction, γ-T3 was reported as the most potent apoptosis inducer on undifferentiated 3T3-L1 preadipocyte [29]. However, Burdeos et al. reported that δ-T3 was the most potent cytotoxicity inducer on differentiated 3T3-L1 adipocytes, followed by γ-T3, β-T3, and α-T3 [27]. This discrepancy in potency was possibly due to the different stage of adipocytes used in cytotoxicity testing. Further study is required to confirm the potency of T3 isoforms in apoptosis induction.

Molecular mechanisms of T3-induced (mainly γ-T3) adipocyte apoptosis was also investigated [28,29]. γ-T3-induced 3T3-L1 preadipocytes apoptosis required 5’-adenosine monophosphate-activated protein kinase (AMPK) signaling, downregulation of B-cell lymphoma-2 (Bcl-2), PPARγ, Akt and extracellular signal-regulated kinase (ERK) signaling, upregulation of Fas ligand and Bcl-2-associated x protein (Bax) expression, and activation of caspase-3 and c-Jun N-terminal kinase (JNK) signaling [29]. γ-T3 induced hASCs apoptosis with increased Bax/Bcl-2 ratio, caspase-3 activation, and karyorrhexis [28]. Molecular ordering analysis further revealed that γ-T3 modulated the nutrient-sensing pathways by activating AMPK but inhibiting Akt signaling in the early phase of treatment [28]. Inhibition on Akt signaling subsequently interfered with mammalian target of rapamycin (mTOR) and its downstream signaling in hASCs [28]. Activation of AMPK signaling is crucial in anti-adipogenic activities of γ-T3, whereby inhibition of the AMPK pathway partially reversed the γ-T3-induced anti-adipogenic effect [28]. In addition, AMPK inhibition did not alter γ-T3-induced autophagy which suggested that autophagy might not directly relate to the anti-adipogenic activity of γ-T3 [28].

### 3.3. Other Beneficial Effects of T3 on Obesity Management

#### 3.3.1. Anti-Inflammatory Activity of T3

Obesity is closely associated with low-grade inflammation, which subsequently increases the risk of cardiovascular diseases, metabolic syndrome, insulin resistance, and diabetes mellitus [42]. Anti-inflammatory properties of T3 were reported in several cell lines and obesity animal models [33,34,35,40,43,44]. T3 in RBEE (5%) [34], muscadine grape seed oil (MGSO) [33], TRF from MGSO [33], annatto oil T3 mixture [40], and γ-T3 [35,41,44] were reported to suppress the release and expression of pro-inflammatory mediators including TNF-α, granulocyte colony-stimulating factor (G-CSF), C/EBPβ, leptin hormone, interleukin-1β (IL-1β), interleukin-6 (IL-6), interleukin-8 (IL-8), inducible nitrite oxide synthase (iNOS), and/or monocyte chemotactic protein-1 (MCP-1). Lower concentrations of γ-T3 (0.024-2.4µM) also resulted in partial suppression of TNF-α-upregulated MCP-1 and IL-6 expression with the restoration of adiponectin and PPARγ expression in differentiated 3T3-L1 cells [41]. Besides, annatto oil T3 mixture was also reported to upregulate anti-inflammatory cytokine interleukin 10 (IL-10) mRNA levels in the adipose tissue of high-fat diet-fed C57BL/6J mice [40]. In addition, annatto oil T3 mixture [40] and γ-T3 (0.05% diet) [35] also significantly reduced macrophage infiltration in the adipose tissue of obese mice. 

Besides, T3, especially γ-T3, also serves as a potent NF-κB inhibitor [43]. γ-T3 is the most potent NF-κB inhibitor compared to other T3 isoforms via electrophoretic mobility shift assay [43]. γ-T3 (5–25 µM) suppressed transforming growth factor β-activated kinase 1 and receptor-interacting protein-induced NF-κB-dependent reporter gene expression in a concentration-dependent manner [43]. Besides, γ-T3 (25 µM) also completely abrogated TNFα-induced NF-κB activation in human lung adenocarcinoma H1299 cells, human embryonic kidney A293 cells, human breast cancer MCF-7 cells, human multiple myeloma U266 cells, and human squamous cell carcinoma SCC4 cells [43]. Furthermore, γ-T3 also abrogated the NF-κB activation by TNFα, phorbol myristate acetate (PMA), okadaic acid, lipopolysaccharide (LPS), cigarette smoke condensate (CSC), IL-1β, and epidermal growth factor (EGF) in human myeloid leukaemia KBM-5 cells [43]. Subsequently, γ-T3 downregulated the NF-κB downstream gene products associated with cancer cell survival, proliferation, invasion and angiogenesis [43]. 

Mechanistically, γ-T3 inhibited NF-κB activation by suppressing the upstream nuclear factor-κB inhibitor type α kinase (IKK) and Akt signaling, which subsequently suppressed the nuclear factor-κB inhibitor type α (IκBα) degradation and NF-κB p65 nuclear translocation [35,41,43,44]. In addition, γ-T3 was reported to suppress mitogen-activated protein kinases (MAPKs) activation including JNK, p38, and ERK in LPS-stimulated hASCs [35]. In contradiction, Wang et al. reported that γ-T3 did not suppress LPS-induced TNF-α, IL-10, cyclooxygenase 2 (COX-2) upregulation, or p38 activation in RAW 264.7 macrophages [44]. This discrepancy may be attributable to the study models used, thus warrant further studies to confirm the role of MAPKs in the anti-inflammatory effects of γ-T3. 

#### 3.3.2. Effects of T3 on Glycemic Status and Insulin Sensitivity in Obesity Models

T3, especially γ-T3 and δ-T3, were demonstrated to improve glycemic control on several in vitro models [35,45] and diet-induced obesity animal models [19,35,36,40]. However, no human study on the effect of T3 on glycemic control in the obese population has been reported. In vitro studies showed that γ-T3 restored the glucose uptake, insulin sensitivity, and Akt signaling in differentiated primary mouse adipocytes [45] and primary hASCs [35]. Palm TRF [36] and δ-T3-enriched palm olein [19] were reported to normalize the fasting blood glucose level and improve the postprandial glucose utilization in high fat-fed rats. Besides, annatto oil T3 mixture (90% δ-T3 and 10% γ-T3) also significantly improved glucose tolerance upon intraperitoneal glucose injection without any alteration of serum insulin level in high fat diet-fed C57BL/6J mice [40]. γ-T3 (0.05% in the diet) also normalized blood fasting glucose and insulin levels as a result of enhanced insulin signaling (Akt and insulin receptor subunit-1 (IRS-1) activation) in high fat diet-fed mice [35]. It is noteworthy that T3 inhibited Akt signaling in undifferentiated 3T3-L1 preadipocytes [23,29,32] but not on differentiated adipocytes [45], differentiated primary hASCs [35], and animal models [35,46]. This biphasic response based on the different stages of adipocyte differentiation should be investigated further. Apart from this, palm T3 treatment (60 mg/kg/day; unknown composition) was demonstrated to downregulate 11β-hydroxysteroid dehydrogenase 1 (steroid hormone metabolism enzyme) expression in mesenteric adipocytes of dexamethasone-treated adrenalectomized rats, which might explain its hypoglycemic effect [38]. 

On the other hand, Betik et al. reported that 50 mg/kg bw/day of T3 supplement (Tocomin^®^) for 10 weeks did not significantly alter the pre- and post-exercise intraperitoneal glucose tolerance and insulin sensitivity in Sprague Dawley rats [39]. Similarly, 4-week subcutaneous administration of Tocomin^®^ did not significantly alter the blood glucose and glycated hemoglobin (HbA1c) levels in high fat-diet rats [21]. The purity and composition in Tocomin^®^ may partly explain the negative findings in these studies compared to the others with more promising results.

#### 3.3.3. The Effects of T3 on Liver and Lipid Profile

Obesity is a strong risk factor for nonalcoholic liver disease due to abnormalities in lipid metabolism and local and systemic inflammation, which altogether impair liver function [47]. T3 was reported to improve liver profile in obesity models [19,36,40] Palm TRF [36], α-T3, γ-T3, or δ-T3-enriched palm olein (85 mg/kg bw/day) [19] were demonstrated to reduce liver injury in diet-induced obese Wistar rats by reducing liver injury markers including plasma aspartate transaminase (AST) and alanine transaminase (ALT) levels. In addition, annatto oil T3 mixture (400 and 1600 mg/kg) also suppressed hepatic steatosis and decreased the liver lipid droplets in diet-induced obese C57BL/6J mice [40]. Annatto oil T3 mixture also significantly reduced liver inflammation of obese mice, indicated by lower TNF-α mRNA level [40]. Parallel with this, TRF, α-T3, γ-T3, and δ-T3 also attenuated hepatic steatosis and inflammatory cells infiltration in the liver of obese rats [19,36]. There is accumulated evidence demonstrating that hepatic PPARα activation is closely related to the reduction of hepatic inflammation [48]. PPARδ activation was also demonstrated to promote anti-inflammatory gene expression [48,49]. Therefore, it is logical to suggest that T3 reduces hepatic inflammation and injury via PPARα/ δ activation. Moreover, T3 also improved blood lipid profile in several obese animal models [19,20,31,36,40]. T3 treatment, including palm TRF [36] (mixture of several T3s and tocopherol isoforms), T3-rich RBO (5 or 10 mg/day) [20], annatto oil T3 mixture [40], and δ-T3-enriched palm olein (85 mg/kg bw/day) [19] significantly reduced blood TG and nonessential fatty acid level in high fat diet-fed rodents. Intraperitoneal injection of γ-T3 (1 mg/day) also reduced blood TG, total cholesterol (TC) and low-density lipoprotein (LDL) level in LDLR–null C57BL/6 mice with no sign of toxicity [31]. 

A beneficial effect of T3 on the lipid and/or liver profiles of the obese population has not been reported so far. Two double-blind placebo-controlled clinical trials were conducted on hypercholesterolemic subjects [31,50]. T3 mixture (120 mg/day; 60 mg for each δ-T3 and γ-T3) managed to reduce TG and chylomicron level on 40 hypercholesterolemic subjects [31]. T3 supplementation (123 mg α-T3, 225.6 mg γ-T3, 51.4 mg δ-T3, and 122.2 mg α-tocopherol) for 1 year caused a marginal improvement on the liver echogenic response and ultrasound examination on these subjects with mild hypercholesterolaemia and nonalcoholic fatty liver [50]. T3 mixture did not improve their glycemic status, lipid profile, liver profile, and renal profile [50]. This study might be limited by the small sample size and further study is required to confirm the findings.

#### 3.3.4. Cardioprotective Effects of T3

Several experiments were conducted to investigate the beneficial effect of Tocomin^®^ [21], palm TRF [36], and purified T3 isoforms [19] in improving the cardiovascular function in obese animal models. Oral administration of palm TRF significantly normalized the systolic blood pressure and improved the noradrenaline-induced thoracic aortic contraction in high-fat-diet-fed rats [36]. TRF also improved systolic function, reduced cardiac eccentric hypertrophy, and diastolic stiffness [36]. Histopathological examination further revealed that TRF inhibited the inflammatory cells infiltration and collagen deposition in the left ventricle of rats [36]. However, palm TRF had no effect on sodium nitroprusside and acetylcholine-induced thoracic aortic relaxation [36]. Similarly, subcutaneous administration of Tocomin^®^ (40 mg/kg/day) also significantly improved Nω-nitro-l-arginine methyl ester (L-NAME, an inhibitor of endothelial nitric oxide synthase (eNOS))-induced contractile response of rat aorta [21]. Tocomin^®^ also improved the sensitivity of acetylcholine but not sodium nitroprusside-induced rat aorta relaxation [21]. Tocomin^®^ induced aortic relaxation via eNOS/soluble guanylate cyclase (sGC)-independent release of endothelial NO [21]. Besides, Tocomin^®^ was demonstrated to restore eNOS and calmodulin (eNOS activator) expressions, Akt signaling and to reduce diet-induced upregulation of caveolin 1 (eNOS suppressor) expression [21]. In addition, Tocomin^®^ also reduced superoxide production in rat aorta via the reduction of nicotinamide adenine dinucleotide phosphate oxidase 2 (Nox2) expression [21]. Moreover, γ-T3- and δ-T3-enriched (but not α-T3) palm olein (85 mg/kg bw/day) improved cardiovascular function by reducing eccentric hypertrophy and normalize systolic blood pressure in diet-induced obese rats [19]. α-T3, γ-T3, and δ-T3 significantly reduced the collagen deposition and inflammatory cell infiltration in the heart muscles [19]. γ-T3 and δ-T3 also improved the thoracic aortic contraction responses to noradrenaline [19]. Contradictory to the previous study [36], γ-T3 and δ-T3 were reported to improve the sodium nitroprusside and acetylcholine-induced relaxation in isolated thoracic aortic rings [19]. Further study is required to confirm these findings. 

Figure 2 summarizes the beneficial effects of T3 on obesity from the current cell culture, preclinical and clinical findings.

## 4. Tocotrienol and Diabetes Mellitus

### 4.1. Antidiabetic Properties of T3 in Cell Lines and Diabetic Animals

Diabetes is closely associated with obesity due to increased insulin resistance and β cell dysfunction as illustrated previously. Several studies concerning the glucose-regulating effects of T3 in obese animal models have been discussed in the previous section. This section will take a closer look at the effects of T3 on diabetes and its complications. 

A multitude of animal studies demonstrated the antidiabetic properties of T3, especially γ-T3 and δ-T3, by preventing diabetic weight loss, hyperphagia, and polydipsia in type-1 diabetes mellitus (T1DM) and T2DM models [45,46,51,52]. Besides, T3 (individual isoform or TRF) significantly improved the glycemic status of diabetic animals by reducing blood glucose level and HbA1c level [45,46,51,52,53,54,55,56,57,58,59,60,61]. In addition, the combination treatment of TRF and insulin resulted in better glycemic control in T1DM rats as compared to single treatment [51,52,57].

Mechanistically, T3 improved glycemic control in diabetic animals by improving insulin synthesis (insulinotropic activity) [45,61] and enhancing insulin sensitivity [45,46,60]. Chia et al. reported that T3 isoforms, including α-T3, γ-T3, and δ-T3, demonstrated insulinotropic activities [61]. δ-T3 is the most potent isoform with insulinotropic activity, followed by γ-T3 and α-T3 [61]. Insulin release genes, including insulin 1 (*INS-1*) and glucose transporter type 2 (*GLUT2*), and insulin gene transcription factors (*PDX1*, *MafA*, and *BETA2*), were upregulated in T3 isoform-treated glucose-stimulated primary normal rat pancreatic β-islet cells [61]. Moreover, cotreatment of T3 isoforms with potassium chloride further enhanced the *INS-1* gene expression [61]. Another similar study by Lee et al. also reported the insulinotropic activities of γ-T3 (1g/kg diet), whereby γ-T3 significantly suppressed the progression of diabetes in BKS.Cg-*Dock*7^m+/+^ Lepr^db^/J mice (*db*/*db* mice) by reducing the fasting blood glucose level and increasing adiponectin level [45]. Besides, γ-T3 also enhanced the clearance of intraperitoneally injected glucose with a two-fold increase of plasma insulin level in diabetic *db*/*db* mice [45]. The histological analysis further revealed that γ-T3 attenuated the loss of pancreatic β-cells and delayed the progression of diabetes in *db/db* mice by increasing the average islet size, size distribution, and insulin-positive area with a lower degree of immune cell infiltration [45]. 

TRF from palm oil (21.8% α-tocopherol, 1.0% γ-tocopherol, 23.4% α-T3, and 37.4% γ-T3) was demonstrated to improve insulin sensitivity by lowering homeostatic model assessment estimated-insulin resistance (HOMA-IR) in high fat-diet and streptozotocin (STZ)-induced diabetic C57BL/6J mice [46]. This effect was attributed to the ability of TRF in restoring the IRS-1 expression, Glut4 expression, and Akt signaling in skeletal muscle of these diabetic mice [46]. Besides, another similar study from Fang et al. also reported that TRF diet (50 mg/kg) improved insulin sensitivity by upregulating PPAR and uncoupling protein 3 (*UCP3*) mRNA level in the muscles of non-fasting T1DM male C57BLKS/J-Lepr Db/Db mice [60]. PPARα was potently activated upon TRF treatment, followed by PPARγ and PPARδ [60]. In addition, purified α-T3 and γ-T3 were reported to serve as a PPARα-selective agonist by activating PPARα [60]. Purified δ-T3, on the other hand, served as pan-PPAR agonist where it greatly activated PPARα and partially activated PPARγ and PPARδ [60]. Furthermore, the molecular analysis also revealed that α-T3, γ-T3, and δ-T3 served as direct PPARα agonist by enhancing the interaction between PPARα and PPARγ coactivator-1α (PGC-1α) in a dose-dependent manner [60]. Parallel with the reporter assay, δ-T3 also demonstrated a greater potency in enhancing PPARα-PGC-1α interaction as compared to others [61]. Molecular docking analysis also demonstrated that δ-T3 exhibited higher affinity and formed greater H-bonding interaction with PPARδ and PPARγ as compared to α-T3, thus explaining its pan-PPAR agonist properties [61]. In addition, α-T3, γ-T3, and δ-T3 were also reported to upregulate *PPARδ* and *PPARγ* mRNA level in glucose-stimulated primary pancreatic β-islet cells in normal rats [61]. Parallel with the molecular studies, δ-T3 did not possess any direct antidiabetic activity where it was reported only weakly reduced the accumulation of AGE products with IC_50_ more than 3 mM in a cell-free anti-AGE assay [62]. Interestingly, T3s, including TRF and the purified isoforms, were reported to downregulate PPARγ in obesity studies but oppositely upregulate and activate PPARγ in diabetic models. We suggested these interesting outcomes may relate to the differential actions of T3 in different stages of disease or types of cells. Parallel with this, the known antidiabetic drug thiazolidinedione (PPARγ agonist) was reported to induce treatment-related fat gain via PPARγ activation [63,64].

On the other hand, several negative findings of T3 on the glycemic control of diabetic animals were reported [62,65,66,67]. Combination of 0.1 g/kg diet T3 mixtures (22.4% α-T3, 1.6% β-T3, 20.8% γ-T3, and 10.1% δ-T3) with astaxanthin (natural carotenoid) did not reduce serum glucose and HbA1c levels in STZ-induced diabetic Osteogenic Disorder Shionogi rats [66]. In addition, Chou et al. demonstrated that RBO diet (15%), which contains 0.9 g of γ-T3, did not significantly alter the plasma glucose and insulin level [67]. Kaup et al. further showed that γ-T3 did not exert any insulinotropic activity in glucose-stimulated INS-1 cells despite very high concentration (100 mg/mL) [65]. These negative findings also partially indicate that only certain T3 isoforms (for instance δ-T3) possess significant antidiabetic activities. 

### 4.2. Antidiabetic Activities of T3 in Human Studies

Findings on the antidiabetic effects of T3 on humans are heterogeneous. The Finnish Mobile Health Examination Survey, with 23-year follow-up, was conducted to investigate the association between dietary intake of T3 isoforms and the risk of T2DM in 2285 men and 2019 women aged 40–69 years old and free from diabetes at baseline [68]. The result showed that only dietary intake of β-T3 was significantly associated with the lower risk of T2DM [68]. The negative finding of other T3 may due to their extremely low dietary intake [68]. Among all T3 isoforms, intake of β-T3 was the highest (2.2 mg/day) [68]. However, this cohort study was limited by its small diabetes case size (383 cases) [68]. Another randomized, double-blind, and placebo-controlled cohort study—Alpha-Tocopherol, Beta-Carotene Cancer Prevention (ATBC) Study—on 29,133 male Finnish smokers (aged 50–69 years old) also reported similar findings [69]. The median follow-up period of this study was 10 years and it had a larger case size (660 cases of T2DM) among 25,505 subjects with complete dietary data [69]. The result showed that only dietary β-T3 was positively associated with lower risk of T2DM [69]. Similar to the earlier study, the dietary intake of β-T3 was the highest among T3 isoforms [69]. This association, however, was found to be insignificant upon adjustment for demographic factors [69]. Nevertheless, this cohort study was limited by recall bias and dietary changes during the study period [69]. 

Several studies showed that supplementation of T3 resulted in better glycemic control on diabetes patients [70,71]. T3-enriched canola oil (200 mg/day) supplied to 45 T2DM patients was demonstrated to reduce their fasting blood glucose level [70]. The randomized controlled trial (RCT) of Vitamin E in Neuroprotection Study (VENUS) was conducted from 2011 until 2015 to identify the effects of T3 on glycemic control and neuroprotection [71]. Among the 229 diabetic patients who completed this trial, oral supplementation of mixed T3 (400 mg/day) for 1 year was shown to improve their glycemic control. However, it did not reduce neuropathic symptoms of diabetes among the patients [71]. The dose of 400 mg T3 per day was considered safe for a human with no observed adverse effect [71].

Wan Nazaimoon et al. supplemented 32 T1DM patients with 1800 mg of a tocotrienol-rich extract, Palmvitee (4% tocopherol and 16% T3 or equivalent to 288 mg T3) or refined palm oil (<0.1% T3) capsules daily for 60 days, followed by a washout period of 60 days and crossed over the treatment for another 60 days [72]. The result showed that both Palmvitee and refined palm oil did not improve the glycemic status of T1DM patients with plateau HbA1c levels [72]. Similarly, a double-blinded and placebo-controlled RCT was conducted on 19 T2DM patients with hyperlipidemia. The subjects received either placebo or a TRF treatment (6 mg/kg body mass/day for 60 days) comprising of 14.6% α-T3, 2.2% β-T3, 38.8% γ-T3, and 2.4% unidentified T3 [73]. The result showed that TRF did not significantly alter the fasting and postprandial glucose level, as well as the glycated hemoglobin level in these T2DM patients [73]. In another RCT, TRF from palm oil (24.5% α-T3, 3.5% β-T3, 35.4% γ-T3, 12.7% δ-T3, and 23.9% α-tocopherol) was orally administrated (552 mg/day) to 86 T2DM patients for eight weeks [74]. It also did not improve the glycemic status of these patients because the HbA1c, serum insulin, plasma glucose levels, and HOMA-IR values were unchanged [74]. The authors suggested that these T2DM patients (both placebo and treatment group) were glycemically stable with a borderline abnormal value of glucose, glycated hemoglobin and serum adiponectin levels [73,74], which might explain the lack of hypoglycemic effect of TRF. 

The effects of T3 in improving diabetes and diabetic-related complications in human studies are summarized in Table 1.

### 4.3. Effects of T3 in Diabetes-Mediated Lipid Abnormalities

Hyperglycemia in T2DM is frequently associated with increased risk of hyperlipidemia, hypercholesterolaemia, hypertriglyceridemia, and hyperinsulinemia [75,76]. These lipid abnormalities are prevalent in diabetes patients due to insulin resistance or alteration of lipid metabolism pathway [75]. Several studies reported that T3 reduced lipid abnormalities in diabetic animals [54,56,59,67]. TRF from palm oil (200 mg/kg bw/day) [54,56] and RBO (400 mg/kg bw/day) [54] were reported to improve lipid profile by restoring the serum high-density lipoprotein cholesterol (HDL-C) level and suppressing the increased serum TC, TG, low-density lipoprotein cholesterol (LDL-C), and very low-density lipoprotein cholesterol (VLDL-C) levels in diabetic rats. Besides, low dose of γ-T3 (0.6 mg/g) in the RBO diet also significantly suppressed hyperlipidemia via reduction of plasma and hepatic TG and LDL-C levels in STZ-induced T2DM rats [59]. Another similar study from Chou et al. demonstrated that high dose of γ-T3 (6 mg/g) in RBO diet also increased HDL-C level and reduced nonesterified fatty acid, liver cholesterol, and TC to HDL-C ratio in T2DM rats [67]. This γ-T3-enriched RBO also significantly increased plasma and liver saturated fatty acid and monounsaturated fatty acid levels with a lower level of polyunsaturated fatty acid [67]. However, this γ-T3-enriched RBO diet did not significantly reduce TC, LDL-C, and TG level [67]. In addition, γ-T3-enriched RBO diet increased faecal neutral sterols and bile acid excretion in diabetic rats [59,67] and upregulated hepatic cholesterol 7α-hydroxylase, LDLR, and HMGCR expression [59]. This is contradictory to the previous studies which reported that T3 downregulated HMGCR expression and activity [77,78]. It is possible that T3 suppression of mevalonate pathway leads to an adaptive response in the liver of the treated rats by increasing HMGCR expression.

TRF was demonstrated to reduce serum total lipids, TC, and LDL-C levels of 19 T2DM patients with hyperlipidemia in one randomized, double-blinded and placebo-controlled trial [73]. However, TRF did not alter the levels of VLDL-C, HDL-C, and TG in these T2DM patients [73]. From human studies, T3-rich extract, Palmvitee (1800 mg/day) was administrated to 32 T1DM patients [72]. Palmvitee did not improve the lipid profile of T1DM patients where the TC, TG, HDL, and LDL levels remained unchanged [72]. In another RCT study, oral administration of TRF from palm oil (552 mg/day) did not improve the lipid profile of 86 T2DM patients, whereby serum TC, TG, HDL-C, LDL-C, and TC/LDL-C ratio were unaltered [74]. This warrants further investigation in confirming the effects of T3 on lipid profile. The negative changes in lipid profile associated with high-dose T3 supplementation suggest that its effects might be U-shaped. 

### 4.4. Anti-Inflammatory Effects of T3 in Diabetic Models

The anti-inflammatory properties of T3 have been investigated extensively [79]. However, there is limited evidence to demonstrate the direct anti-inflammatory action of T3 in diabetic animal or human models. TRF was shown to normalize the inflammation-related markers including NF-κB, MCP-1, IL-6, and TNF-α in the skeletal muscle of STZ-induced diabetic C57BL/6J mice [46]. Besides, γ-T3 was demonstrated to inhibit the downstream activation of caspase-1 and the release of IL-1β and IL-18 in isolated peritoneal macrophages from diabetic *db/db* mice upon stimulation of palmitate (inflammasome activator) and LPS [45]. A similar finding was reported on γ-T3-treated nigericin (inflammasome activator) and LPS-stimulated iJ774 macrophages and primary bone marrow-derived macrophages (BMDMs) from C57BL/6 mice [45]. Mechanistically, suppression of caspase-1 activation and IL-1β secretion by γ-T3 were found to be dependent on nucleotide-binding oligomerization domain-like receptor protein 3 (NLRP3) inflammasome suppression but independent on A20 (a negative regulator of NF-κB) [45]. In addition, anti-inflammatory actions of γ-T3 also involved AMPK signaling. This is evidenced by the almost complete reversal of γ-T3’s suppression of NLRP3 inflammasome activity, caspase-1 activation and IL-1β secretion in palmitate and LPS-stimulated iJ774 macrophages by compound C (AMPK inhibitor) [45]. Moreover, γ-T3 also induced early blockage of NF-κB activation by promoting nonproteasomal degradation of tumor necrosis factor receptor-associated factor 6 (TRAF6; positive regulator of NF-κB) [45]. On the other hand, two double-blinded randomized controlled trials were conducted to investigate the effect of T3 on the inflammatory status of diabetes patients [74,80]. T3-enriched canola oil (200 mg/day) was supplied to 44 T2DM patients for eight weeks [80]. This T3 supplement consists of 13.2% α-T3, 16.6% γ-T3, 8.6% other unspecified T3, and 16% α-tocopherol [80]. The result showed that T3-enriched canola oil significantly reduced serum high sensitivity-C reactive protein (hsCRP) in T2DM patients [80]. In another recent RCT study, oral administration of TRF from palm oil (552 mg/day) did not improve the inflammatory status of these T2DM patients, in which serum inflammatory markers, including hsCRP, IL-6, and TNF-α, were unchanged [74].

### 4.5. Antioxidant Properties of T3 in Diabetic Animal Models

Oxidative–nitrosative stress is involved in the pathogenesis of diabetes mellitus [81,82]. Hyperglycemia was closely related to the upregulation of reactive oxygen species/reactive nitrogen species (ROS/RNS) [83,84]. This oxidative–nitrosative stress subsequently contributes to diabetic cardiovascular disease, retinopathy, neuropathy, and nephropathy [85,86,87]. In this situation, antioxidants including T3 might be beneficial in diabetes management. T3 was demonstrated to reduce oxidative–nitrosative stress in several diabetic animal models as indicated by the reduction of biomarker for oxidative stress including total nitric oxide (NO), malondialdehyde (MDA), and 4-hydroxynonenal (4-HNE) level as well as the restoration of nonthiol-protein expression and antioxidant enzymes activities including superoxide dismutase (SOD), catalase (CAT), glutathione peroxidase (GPx), and glutathione reductase (GR) [46,51,52,53,54,56,57,58,88]. Besides, TRF also reduced oxidative stress markers like 4-HNE and protein carbonyls. These changes occurred concurrently with the reduction of nuclear factor erythroid 2-related factor 2 and heme oxygenase-1 signaling in skeletal muscle of diabetic mice [46]. Muharis et al. also reported that TRF (72% T3; unknown T3 composition) possessed radical scavenging capacity in a dose-dependent manner via 1-diphenyl picrylhydrazyl (DPPH) antioxidant assay [89]. Moreover, the combination of 0.1 g/kg diet T3 mixtures (22.4% α-T3, 1.6% β-T3, 20.8% γ-T3, and 10.1% δ-T3) with 0.1 g/kg diet astaxanthin on STZ-induced diabetic Osteogenic Disorder Shionogi rats significantly reduced the lipid peroxidation in the serum and liver but not the urine 8-hydroxy-2′-deoxyguanosine (8-OHdG; biomarker for oxidative DNA damage) level [66]. Furthermore, T3 also significantly suppressed diabetes-related oxidative DNA damage on leukocytes from diabetic rats [56,58]. In a human study, T3-enriched canola oil (200 mg/day) significantly reduced serum MDA level and increased total antioxidant capacity of 45 T2DM patients [70]. On the other hand, the supplementation of Palmvitee (1800 mg/day) and refined palm oil (<0.1% T3) significantly reduced the MDA level of T1DM patients [72]. However, supplementation of Palmvitee with a higher level of T3 did not further reduce the MDA level, which suggested the antioxidant activity was mostly due to the refined palm oil [72].

### 4.6. Effects of T3 in Diabetic Nephropathy

Nephroprotective effects of T3 were reported as well, whereby TRF from palm oil and RBO [53,54] and T3 mixture [52] were demonstrated to improve the renal function of diabetic animals. The evidence on the nephroprotective effects of T3 among diabetes patients remains scarce. In detail, T3 treatments were shown to increase urea and creatinine clearance, and reduce diabetes-related polyuria, hyperalbuminuria, hypercreatinuria, proteinuria, hypercreatinemia, and high blood urea nitrogen (BUN) levels in diabetic animal models [52,53,54]. Besides, T3 mixture suppressed oxidative–nitrosative stress, inflammation, and cellular damage of kidneys through abolishing STZ-upregulated TNF-α, transforming growth factor-1β, NF-κB, and caspase-3 activities [52]. In addition, histological analysis revealed that both TRF from palm oil and RBO reduced diabetic tubules and glomerular damages, tubulointerstitial inflammation and mesangial expansion in the kidney of diabetic rats [53,54]. There was downregulation of transforming growth factor β, collagen type IV, and fibronectin protein expression with TRF treatments, indicating the improvement of renal fibrosis [54]. Furthermore, the potency of TRF palm oil was almost double compared to TRF RBO [54]. This was attributed to the higher amount of γ-T3 present in TRF palm oil [53,54]. This is supported by the fact that γ-T3 has greater antioxidant activity compared to other T3 isoforms [90]. This further emphasizes the crucial role of antioxidant activity of T3 in suppressing diabetic nephropathy. Lastly, T3-enriched canola oil (200 mg/day) was tested in one double-blinded RCT on 44 T2DM patients [80]. The T3 supplementation significantly reduced urine microalbumin but did not reduce serum NO level, urine creatinine level, and urine volume in these T2DM patients [80]. The lack of positive changes due to T3 supplementation may be dose-related. 

### 4.7. Effects of T3 in Diabetes-Related Cardiovascular Diseases

Several clinical and epidemiological studies have identified that cardiovascular disease is one of the major complications of diabetes mellitus [91,92]. T3 was demonstrated to be beneficial in reducing hypertensive and cardiovascular complications in diabetic animal models [56,89]. TRF treatment (200 mg/kg bw/day; unknown composition) was reported to protect the thoracic aorta from diabetes-mediated vascular wall alterations in STZ-induced diabetic rats with a significant improvement of redox status [56]. Biochemical and histological examination further revealed that TRF also suppressed the vascular smooth muscle cell degeneration in the aorta and inhibited the formation of electron-dense amorphous material in the aortic media [56]. Muharis et al. also reported that TRF (72% T3; unknown composition) was beneficial in hypertension management [89]. TRF incubation significantly enhanced the acetylcholine-induced relaxation of aorta rings in spontaneously hypertensive rats (SHR) and STZ-induced diabetic Wistar Kyoto rats [89]. The TRF-enhanced antihypertensive action could involve the modulation of endogenous NO level, possibly by endothelial nitric oxide synthase activity [89]. The beneficial effects of T3 in reducing cardiovascular diseases among diabetes patient are still uncertain. From two randomized, double-blinded, placebo-controlled trials, TRF did not significantly improve the vascular function and systolic and diastolic pressures among T2DM patients [73,74]. The beneficial effects of T3 on cardiovascular health of diabetes patients remain unclear.

### 4.8. Effects of T3 in Diabetic Cataractogenesis

Diabetes-mediated cataractogenesis may be attributable to the diabetes-induced oxidative–nitrosative stress, which subsequently led to polyol accumulation [82]. To date, only two studies were carried out to investigate the anticataractogenic effect of T3 on diabetic animal models [93,94]. Abdul Nasir et al. prepared a T3 mixture (90% δ-T3 and 10% γ-T3) in microemulsion formulation (TTE) or liposomal formulation (TTL) [93,94]. Result showed that topical application of drop vehicle (10 µL) with 0.02 and 0.03% TTE and TTL marginally but significantly slowed down the progression of cataract in galactose diet-induced diabetic rats as indicated by lower ocular opacity index, normalized total lens protein, soluble lens protein, insoluble lens protein, and soluble to insoluble lens protein ratio [93]. Besides, there is no significant difference between TTE and TTL treatment [93]. In addition to this, 0.2% TTE was reported to be not beneficial in preventing diabetic cataract progression. This was possibly due to the prooxidant activities of T3 mixture in a very high concentration [93]. In the second study, TTE was reported to slow down cataractogenesis in STZ-induced diabetic rats with lower lenticular aldose reductase activity, sorbitol level and lenticular NF-κB activation in the eyes [94]. TTE treatment was shown to normalize the lenticular ATP level, Na^+^/K^+^-ATPase activity, plasma membrane Ca^2+^-ATPase activity, sarcoplasmic/endoplasmic reticulum Ca^2+^-ATPase activity, and calpain 2 activity in the eyes of diabetic rats [94]. Moreover, TTE also improved redox status of diabetic rats with an increase of lens GSH content, CAT activity and SOD activity, and lower lenticular iNOS level, 3-nitrotyrosine (oxidative stress biomarker), and MDA level [93,94]. Topical application of TTE did not prevent STZ-induced weight loss and hyperglycemia in diabetic rats [94]. Collectively, it is suggested that antioxidant properties of T3 were crucial to inhibit diabetic cataractogenesis. 

### 4.9. Effects of T3 in Diabetic Neuropathy

T3 was reported to reduce diabetes-related neuropathy in diabetic rats [51,57]. Oral administration of T3 mixture (25, 50, and 100 mg/kg bw/day; unknown composition) dose-dependently improved the behaviour and memory performances of STZ-induced diabetic rats [57]. This T3 mixture also inhibited the STZ-induced upregulation of acetylcholinesterase activity in the cerebral cortex of diabetic rat [57]. Besides, this T3 mixture also suppressed inflammation and cell death events in several regions of the brains of diabetic rats, attributed to lower expression of TNF-α, IL-1β, NF-κB (p65 subunit), and caspase-3 levels [57]. Furthermore, oral gavage of another T3 mixture (100 mg/kg bw/day; 145.6 mg/g α-T3, 11.4 mg/g β-T3, 174.2 mg/g γ-T3, and 128.2 mg/g δ-T3) also significantly suppressed oxidative–nitrosative stress in the sciatic nerves of STZ-induced diabetic rats [51]. This T3 mixture improved the nociceptive threshold over STZ-induced thermal hyperalgesia, mechanical hyperalgesia and tactile allodynia [51]. Similarly to the earlier study, T3 mixture also dose-dependently suppressed STZ-upregulated plasma TNF-α, plasma IL-1β, plasma transforming growth factor-β, and sciatic nerve caspase-3 expression [51], which might be associated to the reduction of cellular damage, cell death and inflammation in diabetic rat brains and nervous tissues. Besides, the VENUS study showed that oral supplementation of mixed T3 (400 mg/day) for one year did not significantly reduce neuropathic symptoms of diabetes on 229 diabetic patients with diabetic peripheral neuropathy syndromes [71]; since all the studies used T3 mixtures [51,57], the lack of significant neuroprotective effects of T3 in VENUS trial could be due to the difference in T3 composition.

### 4.10. Effects of T3 on Skeletal Muscle of Diabetic Animals

In addition, a study reported the beneficial effects of T3 on skeletal muscle of high fat-diet/STZ-induced diabetic C57BL/6J mice [46]. TRF from palm oil (21.8% α-tocopherol, 1.0% γ-tocopherol, 23.4% α-T3, and 37.4% γ-T3) was dissolved in olive oil and orally administrated to diabetic mice (100 and 300 mg/kg bw) for 12 weeks [46]. The results showed that TRF supplementation (300 mg/kg bw) significantly reduced the diabetes-related muscle atrophy via the suppression of skeletal muscle cell apoptosis [46]. Besides, TRF also reduced oxidative stress and inflammation of skeletal muscle [46]. At the cell signaling level, TRF increased mitochondrial biogenesis by restoring sirtuin 1 (SIRT1) and AMPK signaling in skeletal muscle [46]. 

Figure 3 summarizes the preclinical and clinical findings of T3 on diabetes.

## 5. Metabolites of T3 and Their Potential Metabolic Effects

T3 is metabolized in the liver in a manner similar with tocopherols, whereby it undergoes omega-hydroxylation mediated by CYP4F2 and CYP3A4, followed by beta-oxidation, and conjugation to become water-soluble end products to be excreted though urine or faces [95,96]. In HepG2 cell culture, γ-T3 was degraded to γ-carboxyethyl hydroxychroman (CEHC), γ-carboxydimethyloctenyl hydroxychroman (CMBHC), γ-carboxymethylhexenyl hydroxychroman (CMHenHC), γ-carboxydimethyloctenyl hydroxychroman (γ-CDMOenHC), and carboxydimethyldecadienyl hydroxychroman (CDMD(en)(2)HC), whereas α-T3 was degraded to α-CEHC, α-CMBHC, α-CMHenHC, and α-CDMOenHC [97]. An in vitro study, using A59 cell culture, and an in vivo study using rodent model, showed that γ-tocopherol and γ-T3 were metabolized to 13′-carboxychromanol and sulfated 9′-, 11′-, and 13′-carboxychromanol [98]. The long-chain vitamin E metabolites and CEHC were shown to have important biological activities. In vitro studies showed that α-CEHC reduced TNF-α induced nitrite production in rat aortic endothelial cells and mouse microglial culture. α-CEHC and γ-CECH inhibited LPS-induced nitrite efflux and prostaglandin E-2 production in microglial. These effects were not shared by α- and γ-tocopherol [99]. In neutrophils, α-, γ-, and δ-tocopherol as well as α-, γ-, and δ-CEHC inhibited superoxide anion production induced by phorbol ester. This was achieved by inhibiting the translocation and activation of protein kinase C (PKC). The CEHCs demonstrated stronger inhibitory effects on PKC compared to tocopherols [100]. The presence of α- or γ-CEHC together with a-tocopherol also delayed the oxidation of LDL better than alpha-tocopherol alone [101]. 9′- and 13′-carboxychromanol generated from vitamin E metabolism were proved to be potent inhibitors of cyclooxygenase-2 [102]. 13′-carboxychromanol also suppressed leukotriene B4 production in human blood neutrophils and HL-60 cells as well as inhibiting human recombinant 5-lipoxygenase activity [103]. α-13′-hydroxy- and α-13′-carboxychromanol were also shown to reduce oxidized LDL uptake and lipid accumulation in human macrophages in vitro by suppressing phagocytosis of oxidized LDL [104]. Thus, these metabolites may play an important role in preventing atherosclerosis and the proinflammatory states in obesity and diabetes.

## 6. Difference in the Metabolic Effect between T3 and Alpha-Tocopherol

Despite the promising metabolic effects of T3, α-tocopherol is still the most abundant vitamin E isoform in the body. Pharmacokinetic studies showed that α-tocopherol was distributed more equally in all the lipoprotein fractions and T3 was detected mainly in HDL-cholesterol [105]. Studies suggested that α-tocopherol was secreted mainly in chylomicrons and T3 in HDL small particles. Thus, T3 was shown to be distributed to adipose-rich tissues, like epididymal fat, perirenal fat, and skin, while α-tocopherol was distributed more evenly to all tissues [106]. The presence of the highly selective α-tocopherol transfer protein in the liver dictates the higher bioavailability of α-tocopherol in the body compared to all vitamin E isoforms [107]. 

α-tocopherol exhibits many metabolic activities similar with T3. In terms of adipogenesis, α-tocopherol was shown to increase thermogenic adipocyte differentiation in mammalian white adipose tissues and in 3T3-L1 cell culture [108]. It also suppressed IL-6 production (a proinflammatory cytokine) by 3T3-L1 adipocytes induced by LPS without affecting IL-10 (an anti-inflammatory cytokine) production [109]. However, α-tocopherol failed to alter fat accumulation in preadipocytes and differentiated adipocytes [110]. In addition, Wu et al. also demonstrated that α-tocopherol is 2.5 times less potent in inhibiting the proliferation of 3T3-L1 adipocytes as compared to α-T3 [29]. Wong et al. also reported that δ-T3, but not α-tocopherol, significantly reduced total fat mass, abdominal circumference, adiposity index, and retroperitoneal and epididymal fat pads mass in high carbohydrate/fat diet-fed rats [19]. 

In terms of glucose regulation, α-tocopherol showed promising hypoglycemic effects in animal studies, by increasing insulin-secreting cells in the pancreas and insulin production, while reducing oxidative products and apoptosis in the pancreas [111,112]. α-tocopherol was less potent as compared to T3 mixture in reducing the plasma glucose or insulin levels in diabetic rats [19,52]. However, translation of the hypoglycemic effects of α-tocopherol to humans proves to be difficult. The ATBC Study showed that α-tocopherol supplementation (50 mg/day) for 5–8 years among Finnish male smokers did not alter the incidence of type 2 diabetes (relative risk 0.92 (95% CI 0.79–1.07)) [113]. α-tocopherol (50 mg/day) also did not prevent macrovascular complications (relative risk 0.84 (95% confidence interval (CI) 0.65–1.10)) and total mortality due to diabetes (relative risk 1.00 (95% CI 0.80–1.25)) among this cohort [114]. A randomized, double-blinded, placebo-controlled trial conducted by de Oliveira et al. demonstrated that α-tocopherol supplementation (800 mg/day) alone or in combination with lipoic acid (600mg/day) did not alter the lipid profile, glucose, insulin, or HOMA-IR of type 2 diabetic patients after 16 weeks [115]. Furthermore, a randomized, double-blinded, placebo-controlled trial by Wu et al. showed that supplementation of α-tocopherol (500 mg/kg) or mixed tocopherols (500 mg/kg, 60% γ-tocopherol) for six weeks increased systolic and diastolic blood pressure, pulse pressure and heart rate compared to placebo group in diabetic patients. The plasma F2-isoprostanes were reduced but endothelium-dependent and -independent vasodilation were not altered with both treatments [116]. These two supplementations also did not alter plasma C-reactive protein, IL-6, TNF-α, or MCP-1 in the patients [117]. Only mixed tocopherols decreased stimulated neutrophil leukotriene B4 production [117]. 

The reason underlying the discrepancies on the biological action of α-tocopherol and T3 is not clear. In term of antioxidant activities, the double bonds of T3 enables it to distribute more evenly in lipid membrane and disorganize the lipid membrane structure, allowing more interactions with free radicals. These factors contribute to its higher free radical scavenging capacities and redox recycling efficacy compared to its tocopherol counterpart [118,119]. Maniam et al. showed that palm T3 (100 mg/kg) significantly reduced lipid peroxidation product and increased GPx activity in the bone of normal male rats but α-tocopherol did not show similar effects [120]. Palm T3 (60 mg/kg for two months) also improved the plasma SOD, GPx, and reduced lipid peroxidation product in ovariectomized rats whereas α-tocopherol did not alter the antioxidant status [121]. Similarly, α-tocopherol was less potent in reducing the lipid peroxidation product and NO levels, and restoring nonprotein thiols, SOD, and CAT levels in STZ-induced diabetic rats as compared with T3 mixture [52]. However, the relative antioxidant activity of T3 and α-tocopherol is still debatable because it is largely dependent on the environment and the assay used [122]. With respect to inflammation, T3 also exhibited better anti-inflammatory effects compared to α-tocopherol. T3-enriched fraction and γ-T3 (60 mg/kg for 2 months) prevented the increase in IL-1 and IL-6 level but α-tocopherol at the same dose only lowered IL-1 level in rats administered with nicotine [123]. In another study, although both α-tocopherol and palm T3 (30, 60, and 100 mg/kg for 2 months) reduced circulating IL-6 level, only palm T3 suppressed IL-1 level in male rats administered with ferric nitrilotriacetate [124]. In addition, molecular analysis revealed that α-tocopherol is not a direct agonist for PPARα as it did not enhance the interaction of PPARα with PGC-1α [60]. Besides, γ-T3, but not γ-tocopherol, was reported to inhibit TNFα-induced NFκB activation via electrophoretic mobility shift assay (EMSA) [43]. Furthermore, T3 mixture produced more pronounced effects in comparison to α-tocopherol in reducing the upregulated TNFα, tissue growth factor-1β, NFκB, and caspase-3 levels in the kidneys of STZ-induced diabetic rats [52]. In diet-induced obese rats, δ-T3 significantly improved the heart and liver structures and functions, while α-tocopherol was reported with no or minimal effects [19].

The differential effects of α-tocopherol and T3 on HMGCR and cholesterol synthesis are rather established. Shibata et al. showed that α-tocopherol alone did not possess lipid-lowering effect, but it suppressed the effect of T3. It did this by lowering the hepatic and adipose concentration of T3 [125]. A study in chicken showed that a vitamin E mixture containing >30% α-tocopherol significantly attenuated the HMGCR-suppressing activity of γ-T3 [126]. A study in male hamsters supplemented with isolated corn oil triglycerides and α-tocopherol showed a worse lipid profile marked by raised serum TC, LDL-C, and TG levels. In addition, α-tocopherol at lower dose suppressed (30 ppm diet) HMGCR activity but at higher dose (81 ppm diet) stimulated HMGCR activity. Co-supplementation of T3 (10 mg) and α-tocopherol (5 mg) also attenuated the inhibition of HMGCR activity [127].

## 7. Conclusion and Perspectives

The evidence accumulated so far, mostly preclinical, supports the therapeutic role of high doses of T3 in preventing or reversing obesity and diabetes, and their multisystemic complications. However, it is still far from being used clinically in the management of obese and diabetic patients. Several issues need to be resolved prior to its application among patients. Firstly, most researchers used naturally derived T3 with varying of compositions and purity in their studies. While this is a comprehensible decision due to the high cost involved in extracting individual T3 isoforms from the sources, it complicates cross-comparison of the findings between studies. The current evidence shows that a mixture richer in γT3 like palm T3 (in contrast to T3 from rice bran) seems to demonstrate better effects. Secondly, different T3 dosages have been used in previous studies to assess its anti-obesity and antidiabetic effects. This, together with the heterogeneity of the T3 mixtures used, complicates the determination of T3 dosage in achieving its optimal therapeutic effects. From the human trials conducted, a daily dose of 200 mg has been shown to improve the glycemic status of diabetic subjects partially. It should be noted that high-dose T3 is potentially harmful, despite the lack of direct evidence in humans. Animal studies showed that high-dose T3 could increase bleeding time and induce liver abnormality in animals [128,129]. There is a paucity of human evidence on the antidiabetic effects of T3, although animal studies suggested a dose as low as 60 mg/kg could reduce fat mass in glucocorticoid-treated rats. Thirdly, the molecular mechanism of T3 in augmenting energy expenses of the body is still superficial. Most of the researchers attribute the biological effects of T3 to its antioxidant and anti-inflammatory properties, but studies have shown that T3 can influence cell signaling pathways. T3 can downregulate PPARγ, which is responsible for the differentiation of adipocytes. It can also suppress the mevalonate pathway, thereby reducing cholesterol synthesis. These limitations and research gaps warrant further studies to validate the function of T3 as an agent to combat metabolic disorders in humans. The authors suggest that at the current stage, T3 has the greatest potential to be developed as a functional food to be used in conjunction with current obesity and diabetes management strategies for the benefits of the patients.

## Figures and Tables

**Figure 1 molecules-24-00923-f001:**
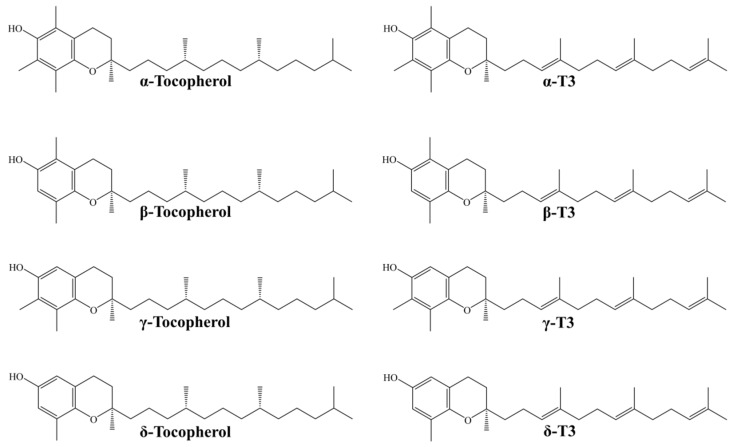
The chemical structure of four tocopherol and tocotrienol isoforms.

**Figure 2 molecules-24-00923-f002:**
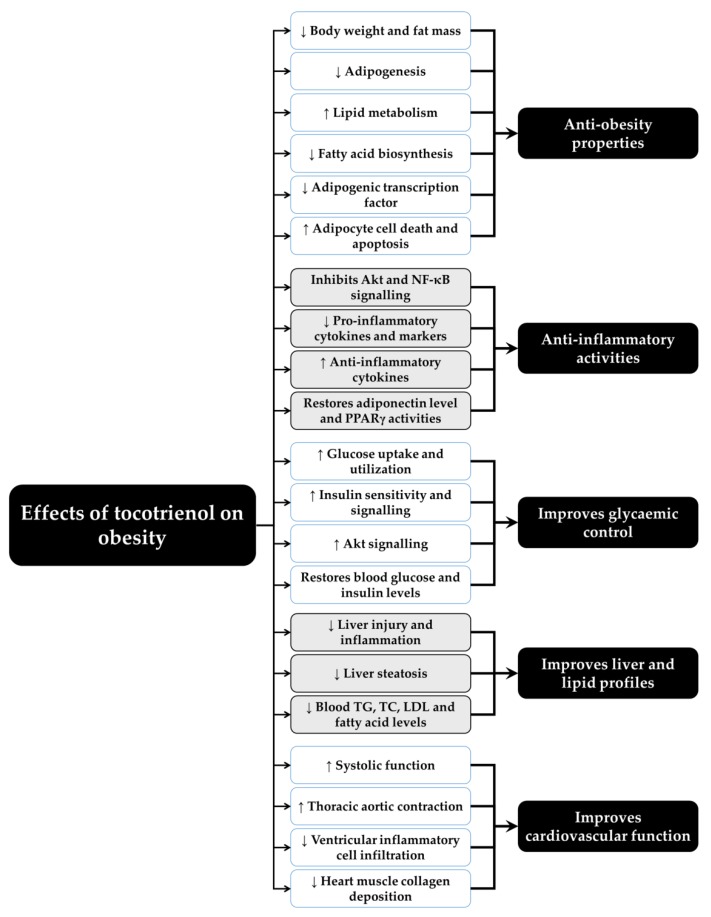
The beneficial effects of tocotrienol on obesity from cell culture, preclinical, and clinical studies. Abbreviations: ↑ = increase or upregulate, ↓ = decrease or downregulate, LDL = low-density lipoprotein, NF-κB = nuclear factor-κB, PPARγ = peroxisome proliferator-activated receptor γ, TG = triglyceride, TC = total cholesterol.

**Figure 3 molecules-24-00923-f003:**
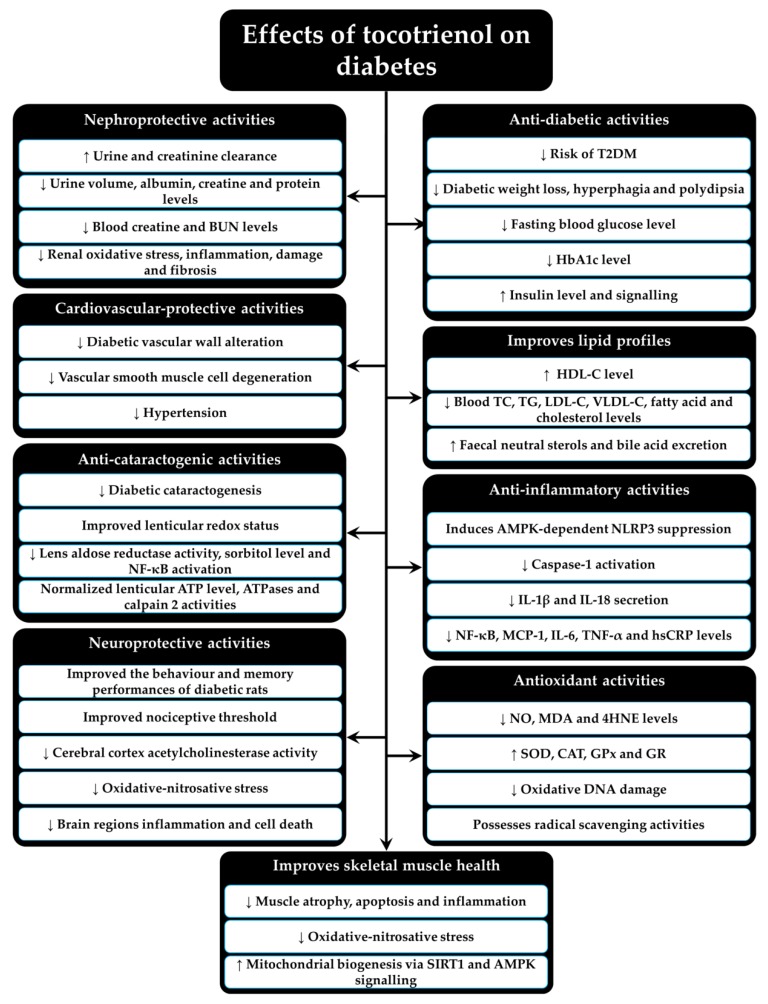
The beneficial effects of tocotrienol on diabetes from preclinical and clinical studies. Abbreviations: ↑ = increase or upregulate, ↓ = decrease or downregulate, AMPK = 5′-adenosine monophosphate-activated protein kinase, BUN = blood urea nitrogen, CAT = catalase, GPx = glutathione peroxidase, GR = glutathione reductase, HbA1c = glycated hemoglobin, HDL-C = high-density lipoprotein, hsCRP= high sensitivity-C reactive protein, 4HNE = 4-hydroxynonenal, IL-1β= interleukin 1β, IL-6= interleukin 6, IL-18= interleukin 18, LDL-C = low-density lipoprotein, MCP-1= monocyte chemotactic protein-1, MDA= malondialdehyde, NF-κB = nuclear factor-κB, NLRP3= nucleotide-binding oligomerization domain-like receptor protein 3, NO = nitric oxide, SIRT1= sirtuin 1, SOD= superoxide dismutase, T2DM= type 2 diabetes mellitus, TC = total cholesterol, TG = triglyceride, TNF-α= tumor necrosis factor-α, VLDL-C = very low-density lipoprotein cholesterol.

**Table 1 molecules-24-00923-t001:** Summary of the antidiabetic effects of T3 from human studies.

T3 Isoform	Treatment Condition and Population	Main Outcomes	References
Dietary intake of T3	The risk of diabetes and dietary intake of T3 was investigated via The Finnish Mobile Health Examination survey (cohort study) on 4504 healthy subjects	Only β-T3 was significantly associated with lower risk of T2DM	[68]
	The risk of diabetes and dietary intake of T3 was investigated via ATBC cohort study on 25,505 healthy subjects	None of the T3 isoforms was associated with lower risk of diabetes after multivariate adjustment	[69]
T3 (unknown T3 composition)	Oral mixed T3 (400 mg/day) capsules were supplied to 229 diabetic patients with diabetic peripheral neuropathy syndromes (VENUS study)	400 mg/day is considered as safe to human Improved glycemic control Failed to reduce diabetic neuropathic symptoms	[71]
Palmvitee (16% T3)	Palmvitee (1800 mg) or refined palm oil capsules were provided to 32 T2DM patients for 60 days, followed by 60 days washout period and then crossed over the supplementation for another 60 days	T3-containing Palmvitee supplement reduced lipid peroxidation but not improving patients’ lipid profile and HbA1c level	[72]
TRF (14.6% α-T3, 2.2% β-T3, 38.8% γ-T3, and 2.4% unidentified T3)	TRF treatment (6 mg/kg bw/day) was supplied to 19 T2DM patients for 60 days (RCT study)	Improved several lipid profile parameters but had no effect on the glycemic status and blood pressure	[73]
TRF (24.5% α-T3, 3.5% β-T3, 35.4% γ-T3, 12.7% δ-T3, and 23.9% α-tocopherol)	552 mg/day of TRF capsules was supplied to 86 T2DM patients with impaired vascular function for 8 weeks (RCT study)	Increased plasma T3 isoforms levels upon supplementation. No effect on inflammation, lipid profiles, glucose metabolism, vascular function, systolic, and diastolic pressure	[74]
T3 mixture (13.2% α-T3, 16.6% γ-T3, 8.6% others T3, and 16% α-tocopherol)	200 mg/day of T3 mixture was supplied to 44 T2DM patients for 8 weeks (RCT study)	Reduced microalbuminuria and hsCRP level Had no effect on serum NO level and renal function	[80]
	200 mg/day of T3 mixture was supplied to 45 T2DM patients for 8 weeks (RCT study)	Reduced fasting blood glucose level Increased total antioxidant capacity and suppressed lipid peroxidation	[70]

Abbreviations: ATBC = Alpha-Tocopherol, Beta-Carotene Cancer Prevention study, HbA1c = glycated hemoglobin A1, NO = nitric oxide, RCT = randomized controlled trial, T2DM = Type 2 diabetes mellitus, T3 = tocotrienol, TRF = tocotrienol-rich fraction, VENUS = The Vitamin E in Neuroprotection Study.

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
