# Peer review of "The Role of Tocotrienol in Protecting Against Metabolic Diseases"

_molecules, 2019, doi:10.3390/molecules24050923_

Round 1

Reviewer 1 Report

In their recent manuscript, Pang and Chin review the role of tocotrienols (T3) in the prevention of metabolic diseases. The authors distinguish between different T3-forms such as gamma-, delta, beta- and alpha-T3. They further investigate the role of T3s in the metabolic syndrome, characterized by obesity, type II diabetes and cardiovascular problems. Their review contains 96 references.

The authors comprehensively summarized the recent literature on tocotrienols and their protective effects regarding the molecular mechanisms lying behind the metabolic syndrome. Still, the reviewer is missing one major consideration regarding the metabolism of T3s. Several authors showed that the metabolic transformation of T3 into long chain metabolites play an essential role for the biological or pharmacological activity. The authors will find corresponding information within some recent publications and reviews listed here: Birringer et al, Natural 6-hydroxy-chromanols and –chromenols: structural diversity, biosynthetic pathways and health implications, RSC Advances, 2018

The reviewer suggests to include a chapter or at least a paragraph commenting on t3 metabolism before publication.

Author Response

Please kindly find the response sheet as per attached. Thank you. 

Reviewer 2 Report

An extensive article with a broad literature review.
The diagrams in Figures 2 and 3 would need to be supplemented with relevant literature references discussed in the text, which would facilitate the analysis and review of literature by readers. 

Author Response

(The authors gave the same response as above.)

Reviewer 3 Report

Pang el al review the role of tocotrienols in protecting against metabolic diseases. They focus on two major topics in metabolic disease, obesity and diabetes, and analyze the current literature in detail. The manuscript is well written and has relevant figures, but it needs some changes, additions, and attention to detail (see below).

Major comments

Whereas the effects of tocotrienols is well reviewed, a paragraph reviewing and comparing the features and effects of tocotrienols with tocopherols should be added: what is similar and what makes them different? Do the tocotrienols affect the same pathways of signal transduction and gene expression as the tocopherols? This is important in view of the fact the only one of the eight vitamin E analogues can reach high concentrations in plasma and tissue (alpha-tocopherol), and thus may have an advantage over all the other analogues. Why are all the other analogues metabolized and eliminated so that they do not reach high levels and are such low levels enough to have any physiological effects?

It would be nice to show a summary table of the few studies of tocotrienols done with human subjects.

Figure 1: usually, the chemical structure is depicted with the chromanols facing to the right, e.g. compare to (1)

Minor comments

Line 22: receptors

Line 26: anti-obesity effects

Line 38: triglycerides levels

Line 73: was shown… is shown

Line 75: Composition: annatto oil is in w/w, whereas the others are in mg/ml

Line 168: might be

Line 173: adipocytes

Line 174: to confirm the potency

Line 175: apoptosis was also

Line 181: with increase of

Line 185: inhibition of the AMPK pathway

Line 187: suggested that autophagy

Line 194: RBEE …explain

Line 254: owing to its antioxidant properties…. Given that alpha-tocopherol has similar antioxidant properties as the tocotrienols (2, 3)but reaches much higher concentration in plasma liver and tissue, shouldn’t it be much better in improving liver profile in obesity models if antioxidant properties are at the basis of this mechanism? Thus, alternative mechanisms should be more discussed for T3 action that are non-antioxidant.

Line 259: droplets

Line 291: L-NAME, an inhibitor of…

Line 296: and to reduce diet-induced

Line 306: summarizes

Line 308: from cell culture, preclinical and clinical studies…

Line 337: mice by increasing

Line 353: demonstrated a greater

Line 423: RBO … explain

Line 436: This is contradictory

Line 443: this sentence is not clear and should be modified

Line 447: this warrants

Line 450: it is unlikely that the small amounts of glycerol in T3 supplements would affect the systemic triglyceride levels

Line 555: lens NF-kB activation…. not clear, do lenses express genes?

Line 608:  therapeutic role of high doses of T3

Line 614: prevents cross-comparison…. complicates cross-comparison

Line 619: daily dose …. mg/daily… daily is repeated

References

1.            Miyazawa T., Burdeos G.C., Itaya M., Nakagawa K., and Miyazawa T. (2019) Vitamin E: Regulatory redox interactions. IUBMB life.

2.            Muller L., Theile K., and Bohm V. (2010) In vitro antioxidant activity of tocopherols and tocotrienols and comparison of vitamin E concentration and lipophilic antioxidant capacity in human plasma. Mol Nutr Food Res 54, 731-742.

3.            Kamal-Eldin A.. and Appelqvist L.A. (1996) The chemistry and antioxidant properties of tocopherols and tocotrienols. Lipids 31, 671-701.

Author Response

Please kindly find the response sheet as per attached. Than you. 
